# Trends in Energy Security Education with a Focus on Renewable and Nonrenewable Sources

**Jarosław Solarz** [1]**, Małgorzata Gawlik-Kobylińska** [2,]*** **, Witold Ostant** [1] **and Paweł Maciejewski** [3]

[1] National Security Faculty, War Studies University, 00-910 Warsaw, Poland; j.solarz@akademia.mil.pl (J.S.);
w.ostant@akademia.mil.pl (W.O.)
[2] Command and Management Faculty, War Studies University, 00-910 Warsaw, Poland
[3] Faculty of Civil Engineering, Czech Technical University in Prague, 166 29 Prague, Czech Republic;
ppmaciej@yahoo.co.uk
* Correspondence: ggawlik2000@yahoo.com

**Abstract:** Energy security education explores various issues, such as a secure and competitive economy and nuclear safety. In the context of energy transition and sustainable development, it also addresses the world's reliance on nonrenewable and renewable energy sources. The aim of this study was to identify research trends pertaining to energy security education, paying particular attention to renewable and nonrenewable sources. This was accomplished with the use of mixed-method research in two steps. The first step was a text-mining and content analysis of publications on energy security education published on the Web of Science platform between 2016 and 2021. From 660 publications on energy security education, titles, abstracts, and keywords were extracted and analysed with NVivo software to identify the most frequent concepts on energy sources in publications. The concepts were associated with nonrenewable energy sources (coal, natural gas, uranium, petroleum, and fossil fuels), nuclear power, and renewable energy sources (hydro, geothermal, solar, tide/wave/ocean, wind, solid biofuels, biogases, liquid biofuels, and renewable municipal waste). The second step was conducting detailed searches with Boolean operators, where "energy security education" was juxtaposed with the distinguished keywords. All searches on energy security education showed that publication activity tended to decrease, while citations increased. The most explored topics concerned: "fossil fuels", "oil, petroleum", "renewable" energy, and "solar" energy sources. An increasing trend was observed for all renewable energy sources as well as selected nonrenewable sources: "oil, petroleum", "nonrenewable", and "coal". Additionally, R-squared values were calculated to indicate the fit of the trendline to the model. Due to the technologically enhanced energy transition and didactic innovations, education focussing on energy sources is expected to remain in demand. Curricula will need to be revised in the future to better reflect this reality.

**Keywords:** energy security education; renewable sources; nonrenewable sources

## 1. Introduction

Since energy security has become a major global challenge of the 21st century [1–3], education on this topic has a significant role in implementing solutions. It fosters the realisation of the transnational goals on the rational use of energy sources and the implementation of alternative energy sources. "Energy security education" is a combination of two different subjects: "security education" and "energy education". The first subject, "security education" [4], aims to furnish students with competencies to combat threats of different natures [5], whereas "energy education" is "one of the ways to make society more aware and active while taking actions towards rational energy use" [6], or an "interdisciplinary approach to teaching and learning about energy" [7]. The necessity of "energy security education" is stressed in publications concerning local perspectives in the context of national security [8] and sustainable development [9]. The literature does not provide a universal definition of energy security education but provides examples of didactic methods, mostly

case studies [10], simulations, serious games [11], tools, and learning environments. It can be noted that this concept appears in NATO training initiatives, such as the NATO Energy Security Centre of Excellence [12]. In these documents, energy security education is defined through the scope of the curriculum, which encompasses energy developments and vulnerabilities, the influence of energy, energy's strategic position, national and international competencies, geopolitics, logistical considerations, new technologies, and international cooperation. All of the issues apply to new security challenges and aim to develop a shared understanding of NATO's energy security agenda [13].

Due to different foci, layers, and the overall complexity of energy security education, this article defines it as the process of teaching about contemporary challenges related to energy security on local, regional, and global scales. One of its topics related to global challenges is the transition to low-carbon energy sources. The literature does not provide analyses concerning fundamental interests and trends in energy security education. Therefore, this study aimed to investigate the research directions in energy security education with a particular focus on renewable and nonrenewable energy sources.

Based on the literature review and NVivo Word Frequency Query, topics related to energy sources that are predominant and shape research trends were identified and analysed from a temporal perspective. The search focussed on both renewable and nonrenewable sources of energy. The following research question (RQ) was proposed:

RQ: In the context of renewable and nonrenewable energy sources, what are the research trends in energy security education?

The corresponding hypothesis (H) was formulated:

H: Energy sources still vary across the world [14], and energy security education trends will focus on the use of renewable sources of energy. This is due to energy transition [15], new technologies for green development, and sustainable solutions [16].

To answer the research question and test the hypothesis, mixed-method research was performed: content analysis of Web of Science publications and text mining with NVivo software.

Since the "energy security education" concept encompasses many issues, the search procedure focused on specific energy sources and strict rules. By focussing on energy source types, this theoretical research may contribute to filling the existing research gap concerning energy security education directions and serve as a starting point for further analyses in this field.

## 2. Energy Security and Education

Energy security provides the content for educational endeavours. The energy security concept was created during the oil crisis of the 1970s, which precipitated an awareness of energy security and triggered the creation of the International Energy Agency (IEA) [17]. "Energy security" can be analysed within five dimensions: availability, affordability, accessibility, acceptability, and developability [18]. The ongoing disputes involve natural and human-induced environmental hazards that affect societies [19,20], modernisation of the grid by shifting towards renewables [3], critical infrastructure protection [21], cybersecurity [22], and energy mix and local perspectives [23]. In addition, "energy security" is one of the core interests of the European Union policy. The proposal for a directive emphasises the European Green Deal (EGD), which "establishes the objective of becoming climate neutral in 2050 in a manner that contributes to the European economy, growth and jobs" [24]. The interest is also reflected in the announcement of EU projects such as the Energy Transition Expertise Centre to identify relevant future topics for the energy transition [25]. Additionally, with the Energy Efficiency Directive, the European Parliament "confirmed its support for education schemes for European citizens to learn about energy conservation and efficiency" [26]. Regarding the future of energy security, the challenges will be the transformation of the global energy sector related to the use of nonrenewable and renewable energy sources: the transition to low-carbon energy sources, the development of

robust infrastructure, digitalisation and artificial intelligence in energy systems, and grid resilience [27].

Education on energy security challenges is one of the transformation conditions. An important topic in energy security education relates to nonrenewable sources, such as hard coal, natural gas, uranium, and petroleum [23], and renewable energy sources, such as hydro, geothermal, solar, tide/wave/ocean, wind, solid biofuels, biogases, liquid biofuels, and renewable municipal waste [28]. Starting from nonrenewable sources of energy, the challenges for educational practitioners concern the depletion of fossil fuels and their replacement with renewable energy [29,30], the decline in coal demand, the problem of miners' livelihood vulnerability [31], heating homes with coal-fired furnaces [32], the crucial role of science and education in the digital modernisation of the oil and gas industry [33], building up competence in nuclear energy through training and education, and international and regional cooperation [34]. For renewables, the educational context involves the creation of facilities (Smart Grid Remote Laboratory) that aid in teaching and learning about sustainable energy sources (solar and wind sources) [35], public education and awareness on renewable energy (from the Sun, wind, and water) to help policymakers make decisions [36], and a "next generation" lab (the Renewable Energy Laboratory) that operates under a new experimental teaching method [30]. In the context of renewables, there are complex energy-saving programmes concerning the development of global energy for the post-carbon-based economy [37]. In higher education, an alternative energy-oriented curriculum is recommended to produce a generation fond of nature and explore the environment in line with technology and science [38]. Such programmes help in reducing the asymmetry in energy use between regions [39]. A complex and interdisciplinary approach to teaching energy security education provides a curriculum that integrates literacy and social concepts with science, technology, engineering, and mathematics (STEM) concepts [29]. This intentional integration is valued because it promotes the idea of a green economy and the security of energy supply [40]. Other exemplary activities concerning renewables and energy security education refer to innovative learning environments and tools such as the Learning Platform, BioprotecENV, clinical organic farming that leads a sustainable living [41], or an educational algorithm to train renewable energy technicians [42]. It can be observed that energy security education that focusses on renewable and nonrenewable sources concerns social problems, modernisation, cooperation, the content of curricula, and learning environments and tools. A vast majority of publications refer to renewable sources, and they stress strategies for energy transitions with security implications as well as the roles of renewable and nonrenewable sources in the development of a sustainable energy system. The adverse effects of exploiting nonrenewable sources are embedded within the context of energy system transition. Education is treated as a tool for the transition and guarantee of a secure change.

## 3. Materials and Methods

The materials used in this study were publications from all 10 databases available on the Web of Science platform (online database): Web of Science (WOS), BIOSIS Citation Index (BCI), Current Contents Connect (CCC), Data Citation Index (DRCI), Derwent Innovations Index (DIIDW), KCI-Korean Journal Database (KJD), MEDLINE, Russian Science Citation Index (RSCI), SciELO Citation Index (SCIELO), and Zoological Record (ZOOREC) [43]. The analysis was conducted between 1 and 5 September 2021 and applied to the period between 2017 and 2021 (as of 31 December).

Two phases of searches were performed: the first was to observe the general trend of energy security education in the literature, and the second was to investigate interest in energy security education with a focus on renewable and nonrenewable energy sources. In the searches, publication activity and citations served as trend indicators. The analysis of publication activity was based on the number of publications in a given year, and a citation count analysis was based on the number of citations of recent scientific publications.

It should be mentioned that within the six-year span, the cited reference may refer to a document from a publication outside of the timespan.

### 3.1. Phase One

In the first phase, 660 publications were identified with the basic search: TS = (energy security educ*) and screened for irrelevant topics. No records were removed. This search was conducted in order to identify a general trend in energy security education. R-squared ($R^2$) was calculated and trendlines were approximated. R-squared is an indicator of how well the data fit the regression model. R-squared takes values from 0 to 1. A score closer to zero means that the factor has relatively little effect on the output, and a score close to 1 indicates that the regression prediction fits the data. In this study, the level of acceptance is above 0.9. The trendlines were approximated with different functions, but finally, the polynomial function was chosen, because it best fit the research data. The polynomial equation calculates the least-squares fit through points with the use of the equation: $y = b + c1 + c2*x^2 + c3*x^3 \ldots c6*x^6$, where b and c1 ... c6 are constants.

The extracted publications provided a background for more detailed analyses concerning the types of energy sources. From the 660 publications, abstracts, titles, and keywords were extracted. They were exported as a plain text file to NVivo software, which allows for a deep dive into the data, irrespective of volume and character [44]. Then, word frequency was determined with this software. In this way, keywords applicable to energy source types were identified. The map of keywords created in NVivo is presented in Figure 1.

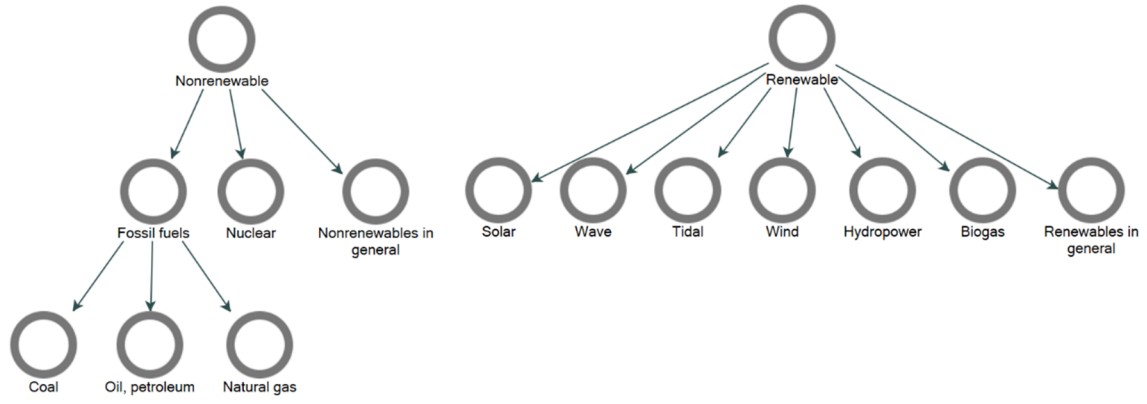

**Figure 1.** NVivo map of keywords concerning nonrenewable and renewable energy sources.

For nonrenewable sources of energy, the following keywords were identified: fossil fuels (coal, oil/petroleum, and natural gas), nuclear power, and nonrenewables in general (texts that use the word "nonrenewable" or "nonrenewables"). Renewable sources were solar energy, wave, tidal, wind power, hydropower, biogas, and renewables in general (texts that use the word "renewable" or "renewables"). In the query, a list of stop words was created, which contained words irrelevant to the study (such as: because, only, such, and that). The listed concepts were used to model the topic of energy security education and indicated the scope of the study.

### 3.2. Phase Two

Having the results of the NVivo analysis, another, more complex search in the Web of Science database was performed with the use of Boolean operators (with the connectors AND, OR). Boolean operators rely on logic, which is a theory of mathematics according to which all variables are either "true" or "false", or "on" or "off". The search relied on three patterns of the "Topic Search" (TS), depending on specific conditions:

(1)  If the concepts from NVivo analysis appear in a singular form ("coal", "petroleum"/ "oil", "natural gas", "nuclear power") and renewable sources ("solar" energy, "wave",

"tidal", "wind" power, "hydropower", "biogas", "geothermal"), the following string was applied: TS = ((a concept from NVivo analysis) AND energy security educ*).

(2) If the concepts were expected to be found in a plural form ("nonrenewable(s)" or "renewable(s)"), the string was: TS = ((a concept from NVivo analysis*) AND energy security educ*). In the query, the asterisk (*) represents any group of characters, including no character.

(3) If the search relied on synonyms (oil and petroleum), the used pattern was: TS = ((a concept from NVivo analysis OR concept from NVivo analysis) AND energy security educ*). This detailed search allowed for gaining insight into energy security education in the context of the types of renewable and nonrenewable energy sources.

The obtained materials enabled the analysis (per year) of publication activity and citations. The results were analysed with R-squared ($R^2$), and trend lines were approximated in the same manner as in Phase One.

The whole procedure allowed for the identification of research trends in energy security education. A research trend was regarded as "the collective action of a group of researchers, each of which begins to pay considerable attention to a specific scientific topic" [45].

The overall research design is presented in Figure 2.

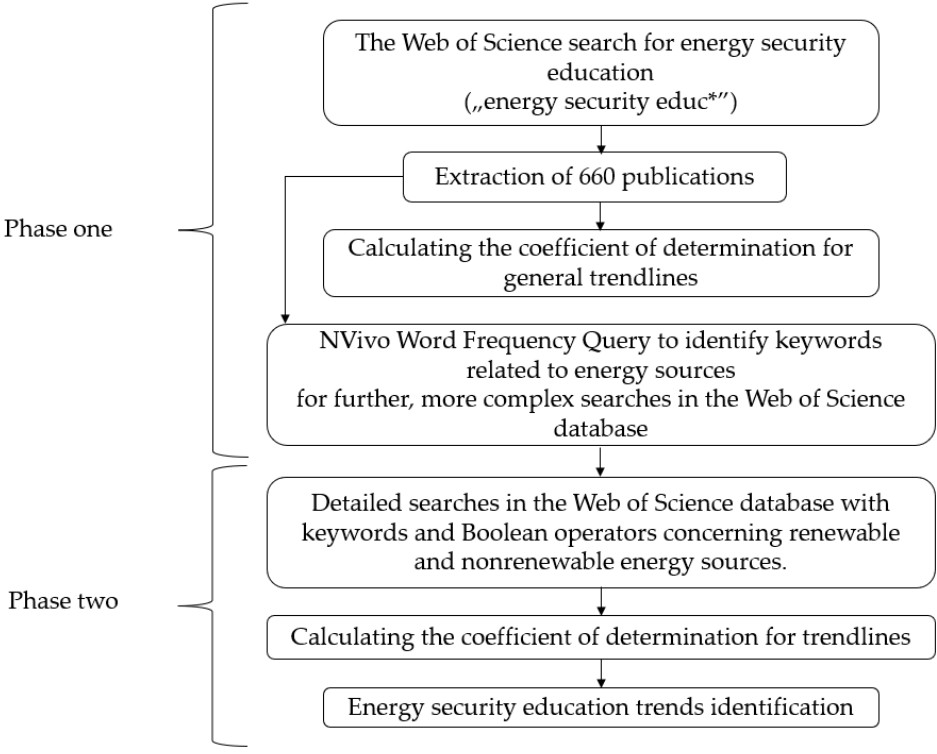

**Figure 2.** Research design for a text-mining and content analysis of publications on energy security education (Web of Science, 2016–2021).

## 4. Limitations

The study had limitations regarding the focus and types of energy sources, the pattern of queries, the timeframe, the searched database, and the interpretation of research results.

The focus on nonrenewable and renewable sources is one of the aspects of energy security education. This topic was chosen due to the scarcity of analyses concerning energy security education with a focus on renewable and nonrenewable energy sources. Regarding the energy type, to the group of nonrenewable energy sources was added "nuclear power", which might be a point of discussion. It should be noted that a nuclear power plant produces renewable energy, but the fuel it uses is not renewable. Nuclear energy was therefore

considered another nonrenewable energy source. Additionally, the study did not include the demand side of energy, such as consumption of households, transport, and industry.

Another limitation concerns the patterns of queries and the choice of keywords, such as "energy security education". The literature presents a lot of variations of energy security topics (such as energy "security awareness education" [46] or "energy education" [6]), but a narrower approach—the use of "energy security education" as one phrase—was adopted to obtained more precise results.

Further limitations apply to the timeframe (from 2016 to 2021) and the Web of Science database (all databases). The choice of the period is related to the fact that the study focused on the most recent activities. The Web of Science platform consists of several literature search databases, which can be effectively searched. As the world's leading citation database, it provides access to publications from the highest-impact journals worldwide.

It should be noted that in the interpretation of the research results, there is a need to consider the time of indexation procedures of publications as well as the ages of papers: it can be expected that older papers have a greater chance of being cited, although this is not a rule. The results should therefore be interpreted with caution.

## 5. Results of Queries in Web of Science Database

The identification of trends relied on observing publication activity and citations. For both indicators, trendlines were added, and the quality of the model was described by R-squared.

### 5.1. Phase One

The results in this phase concern the publication activity and citations referring to energy security education. The first query applied to the general term "energy security education" was run with the following parameters in the basic search: TS = (energy security educ*), which resulted in 660 records. The total number of publications and citation count (without self-citations) per year are presented in Figure 3.

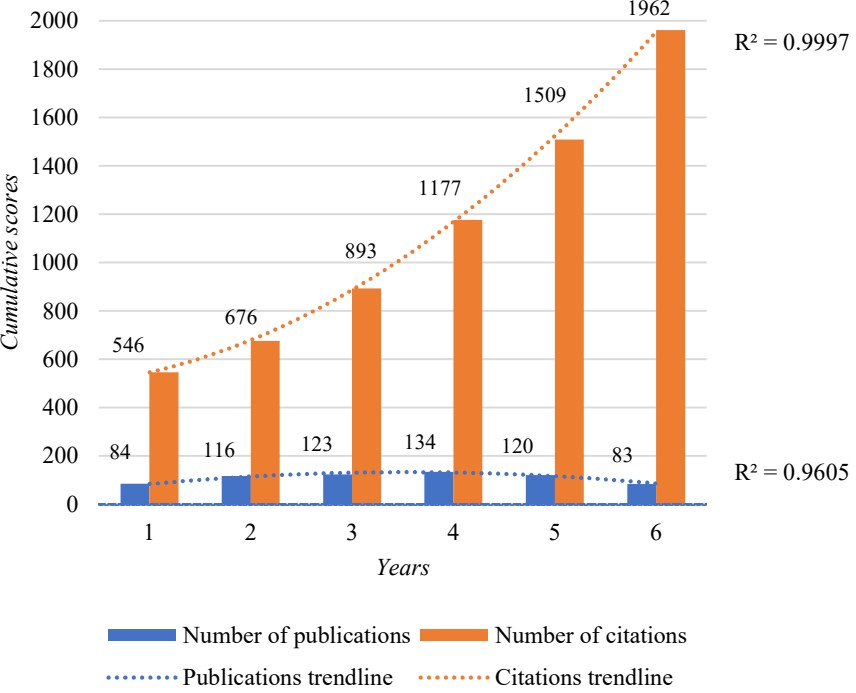

**Figure 3.** Publications and citations based on analysis pattern: topic: TS = (energy security educ*) in 2016–2021. Cumulative scores from the Web of Science database with polynomial trendlines.

The results indicated that in the given period, the number of publications oscillated between 83 and 134 and was the highest in 2019 (134). Then, there was a slight decrease (in 2021—83 publications). The citation count significantly increased year by year. For both indicators, trendlines were approximated. The indicator of a good fit of a model (trendline) is the value of R-squared. The trendline was approximated by the polynomial function. $R^2 = 0.9996$ for publications and $R^2 = 0.9997$ for citations.

*5.2. Phase Two*

With NVivo, keywords were identified that allowed for more complex analyses of energy security education. They included the types of nonrenewable sources ("coal", "petroleum"/"oil", "natural gas", "nuclear power", and "nonrenewables" in general) and renewable sources ("solar" energy, "wave", "tidal", "wind" power, "hydropower", "biogas", "geothermal" energy, and "renewables" in general). Some searches (less than 10 items) were not included in the analysis due to a small amount of data.

5.2.1. Nonrenewable Energy Keywords

- Energy security education concerning nonrenewable energy treated as a general term

The search was performed with the following pattern: TS = ((non renewable*) AND energy security educ*). The keywords included the terms "nonrenewable", "non renewable", and "non-renewable". Only 13 records and 249 citations were identified from 2016 to 2021 (Table 1).

**Table 1.** Publication and citation counts based on the analysis pattern: TS = ((non renewable*) AND energy security educ*). Source: Web of Science database.

| Year | 2016 | 2017 | 2018 | 2019 | 2020 | 2021 | Total |
|---|---|---|---|---|---|---|---|
| Publications | 0 | 3 | 2 | 5 | 2 | 1 | 13 |
| Citations | 41 | 40 | 50 | 45 | 54 | 64 | 249 |

The query results indicate that nonrenewable energy is not a frequent topic of research; however, the increasing number of citations suggests that the topics may appear in other areas of study.

- Energy security education concerning fossil fuels

The query: TS = ((fossil fuels) AND energy security educ*) resulted in 18 publications and 649 citations (Table 2).

**Table 2.** Publication and citation counts based on the analysis pattern: TS = ((fossil fuels) AND energy security educ*). Source: Web of Science database.

| Year | 2016 | 2017 | 2018 | 2019 | 2020 | 2021 | Total |
|---|---|---|---|---|---|---|---|
| Publications | 2 | 8 | 2 | 3 | 2 | 1 | 18 |
| Citations | 80 | 101 | 115 | 101 | 134 | 118 | 649 |

The lower number of publications does not correspond to an increasing number of citations. The descriptions of studies may be vital in different areas of research.

- Energy security education concerning coal

The basic search involving the search: TS = ((coal) AND energy security educ*) resulted in 10 records and 909 citations (Table 3).

**Table 3.** Publication and citation counts based on the analysis pattern: TS = ((coal) AND energy security educ*). Source: Web of Science database.

| Year | 2016 | 2017 | 2018 | 2019 | 2020 | 2021 | Total |
|---|---|---|---|---|---|---|---|
| Publications | 1 | 3 | 1 | 2 | 2 | 1 | 10 |
| Citations | 0 | 7 | 6 | 23 | 27 | 36 | 99 |

It can be noticed that the interest in energy security education in the context of coal topics has been minimal. Additionally, the relatively low number of citations may prove that this combination is not explored in other areas.

- Energy security education concerning petroleum or oil

The query was run with the following parameters: TS = ((oil OR petroleum) AND energy security educ*); it resulted in 26 records and 558 citations (Table 4).

**Table 4.** Publication and citation counts based on the analysis pattern: TS = ((oil OR petroleum) AND energy security educ*). Source: Web of Science database.

| Year | 2016 | 2017 | 2018 | 2019 | 2020 | 2021 | Total |
|---|---|---|---|---|---|---|---|
| Publications | 5 | 9 | 4 | 2 | 3 | 3 | 26 |
| Citations | 60 | 71 | 114 | 85 | 113 | 115 | 558 |

In this case, there is a relatively low number of publications, but the number of citations is growing.

- Energy security education concerning natural gas

The search involved the query run with the following parameters: TS = ((natural gas) AND energy security educ*). The query resulted in 25 records and 1390 citations (Table 5).

**Table 5.** Publication and citation counts based on the analysis pattern: TS = ((natural gas) AND energy security educ*). Source: Web of Science database.

| Year | 2016 | 2017 | 2018 | 2019 | 2020 | 2021 | Total |
|---|---|---|---|---|---|---|---|
| Publications | 3 | 7 | 7 | 4 | 3 | 1 | 25 |
| Citations | 168 | 220 | 188 | 205 | 281 | 328 | 1390 |

Despite the decreasing number of publications, the term "natural gas" in energy security education has gained interest, indicated by the growing citation number.

- Energy security education concerning nuclear power

The search relied on parameters: TS = ((nuclear power) AND energy security educ*). This query resulted in 19 records and 191 citations (Table 6).

**Table 6.** Publication and citation counts based on the analysis pattern: TS = ((nuclear power) AND energy security educ*). Source: Web of Science database.

| Year | 2016 | 2017 | 2018 | 2019 | 2020 | 2021 | Total |
|---|---|---|---|---|---|---|---|
| Publications | 3 | 6 | 2 | 4 | 1 | 2 | 19 |
| Citations | 8 | 8 | 26 | 44 | 61 | 44 | 191 |

Within a relatively low number of publications, a growing number of citations may suggest that areas other than energy security education are explored.

- Energy security education in the context of nonrenewables: an analysis

The results concerning publications and citations on all nonrenewable energy sources in the context of energy security education are visualised in Figures 4 and 5. They also show trendlines (best-fit models).

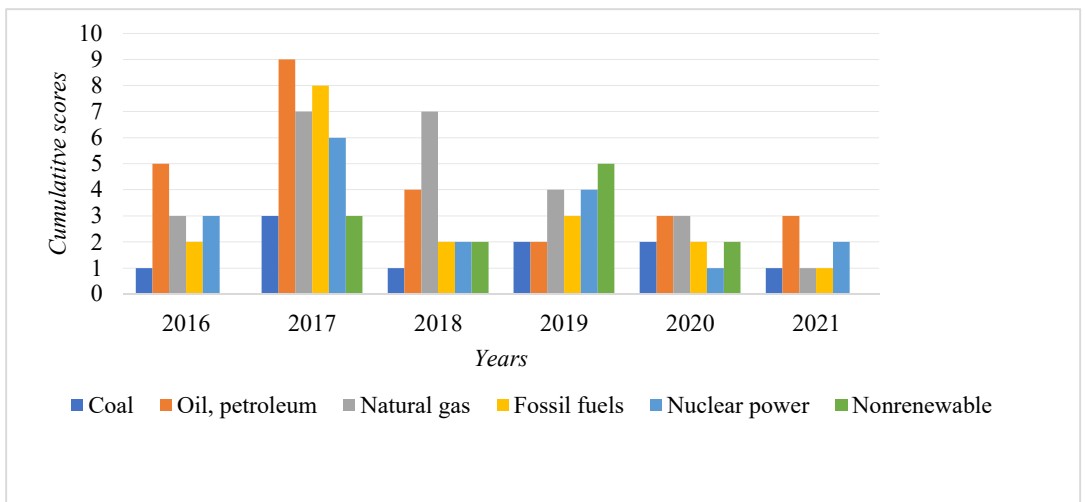

**Figure 4.** Publications on nonrenewable energy sources in the context of energy security education with polynomial trendlines.

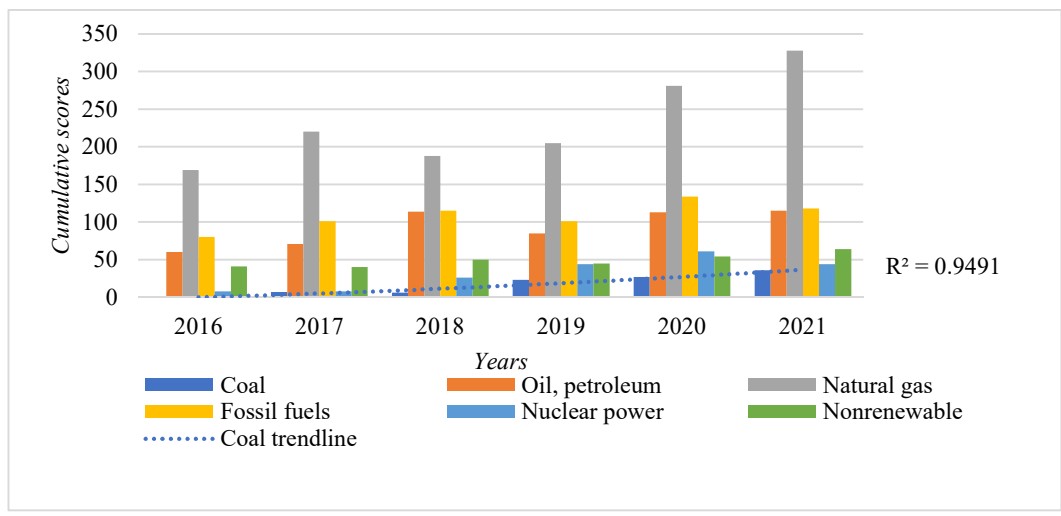

**Figure 5.** Citations on nonrenewable energy sources in the context of energy security education with trendlines.

It can be observed that the number of publications related to nonrenewable sources of energy in the context of energy security education is decreasing for "natural gas", "fossil fuels", and "nonrenewable", while for "oil, petroleum" and "nuclear power" it tends to increase. For all items, R-squared values are below 0.9, which indicates that the line does not fit as well to the data: "natural gas" $R^2 = 0.878$), "nonrenewable" ($R^2 = 0.7202$), "nuclear power" ($R^2 = 0.7284$), "oil, petroleum" ($R^2 = 0.4164$), "fossil fuel" ($R^2 = 0.2998$), and "coal" ($R^2 = 0.1921$). For this reason, trends are not displayed in Figure 4.

Regarding the citations, there was an increase in "oil, petroleum", "nonrenewable", and "coal", while for "fossil fuels" and "nuclear power", there was a decrease in 2021. The most explored topics, expressed as the highest number of citations, concerned "fossil fuels" and "oil, petroleum". Best-fit models concerned "coal" ($R^2 = 0.9491$), while less-fit models applied to "fossil fuels" ($R^2 = 0.8787$), "nonrenewable" energy sources ($R^2 = 0.8763$), "nuclear power" ($R^2 = 0.8266$), "oil, petroleum" ($R^2 = 0.7117$), and "natural gas" ($R^2 = 0.8763$).

5.2.2. Renewable Energy Keywords

- Energy security education concerning renewable energy treated as a general term

Similar search parameters were used in this area: TS = ((renewable*) AND energy security educ*). The query resulted in 93 records and 1712 citations (Table 7).

**Table 7.** Publication and citation counts based on the analysis pattern: TS = ((renewable*) AND energy security educ*). Source: Web of Science database.

| Year | 2016 | 2017 | 2018 | 2019 | 2020 | 2021 | Total |
|---|---|---|---|---|---|---|---|
| Publications | 14 | 25 | 13 | 17 | 14 | 10 | 93 |
| Citations | 246 | 290 | 330 | 367 | 479 | 479 | 1712 |

The number of citations indicates that the context of renewable energy is significant in works related to energy security education combined with the term "renewable", which is generic in nature and may apply to several or all nonrenewable energy sources.

Apart from the general term "renewable", searches were conducted for "geothermal", "biogas", and "hydropower" terms. The following queries resulted in a relatively low number of publications:

TS = ((geothermal) AND energy security educ*): one record (2019—1; 2021—1);

TS = ((biogas) AND energy security educ*): two records (2019—1; 2021—1);

TS = ((hydropower) AND energy security educ*): seven records (2016—1; 2017—2; 2019—2; 2020—1; 2021—1);

TS = ((tidal) AND energy security educ*): three results (2016—1; 2019—1; 2021—1).

These results were not included in the analysis.

- Energy security education concerning solar wave source

With the pattern search TS = ((wave) AND energy security educ*), 19 publications and 107 citations were obtained (Table 8).

**Table 8.** Publication and citation counts based on the analysis pattern: TS = ((wave) AND energy security educ*). Source: Web of Science database.

| Year | 2016 | 2017 | 2018 | 2019 | 2020 | 2021 | Total |
|---|---|---|---|---|---|---|---|
| Publications | 3 | 2 | 2 | 6 | 4 | 2 | 19 |
| Citations | 7 | 6 | 13 | 19 | 20 | 42 | 107 |

There is a relatively low number of publications, but the number of citations seems to increase.

- Energy security education concerning the solar source

The applied search pattern was the following: TS = ((solar) AND energy security educ*), which resulted in 43 publication records and 382 citations (Table 9).

**Table 9.** Publication and citation counts based on analysis of the pattern: TS = ((solar) AND energy security educ*). Source: Web of Science database.

| Year | 2016 | 2017 | 2018 | 2019 | 2020 | 2021 | Total |
|---|---|---|---|---|---|---|---|
| Publications | 9 | 8 | 8 | 7 | 4 | 7 | 43 |
| Citations | 25 | 31 | 38 | 72 | 86 | 127 | 382 |

The number of publications on the solar energy source in energy security education is nonlinear; it tends to increase. However, the number of citations significantly increased, indicating the relevance of this topic for contemporary studies.

- Energy security education concerning wind

The query was based on: TS = ((wind) AND energy security educ*), which resulted in 32 publications and 213 citations (Table 10).

**Table 10.** Publication and citation counts based on the analysis pattern: TS = ((wind) AND energy security educ*). Source: Web of Science database.

| Year | 2016 | 2017 | 2018 | 2019 | 2020 | 2021 | Total |
|---|---|---|---|---|---|---|---|
| Publications | 2 | 5 | 4 | 8 | 7 | 3 | 29 |
| Citations | 24 | 16 | 18 | 50 | 50 | 55 | 213 |

A different number of publications per year and increasing citations may indicate that wind energy in energy security education is receiving increasing interest among scholars.

- Energy security education in the context of renewables: an analysis

The numbers of publications and citations are juxtaposed to verify the research trends in energy security education (Figures 6 and 7).

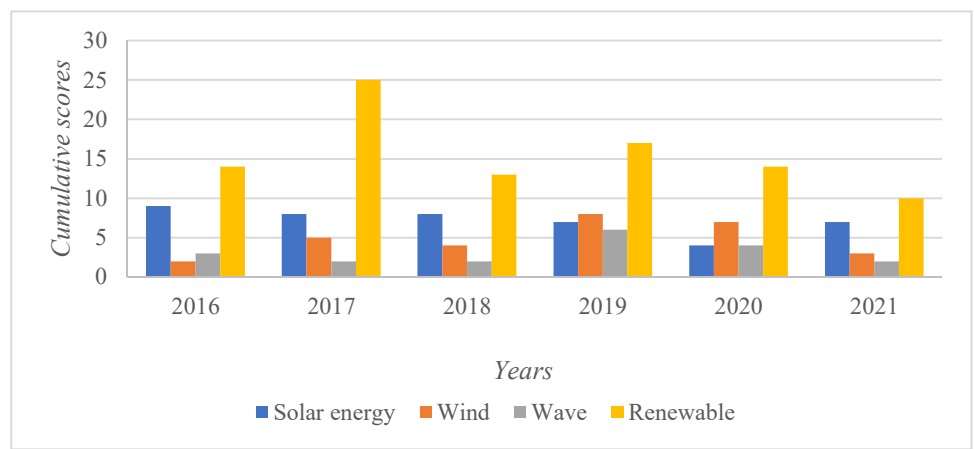

**Figure 6.** Publications on renewable sources in the context of energy security education.

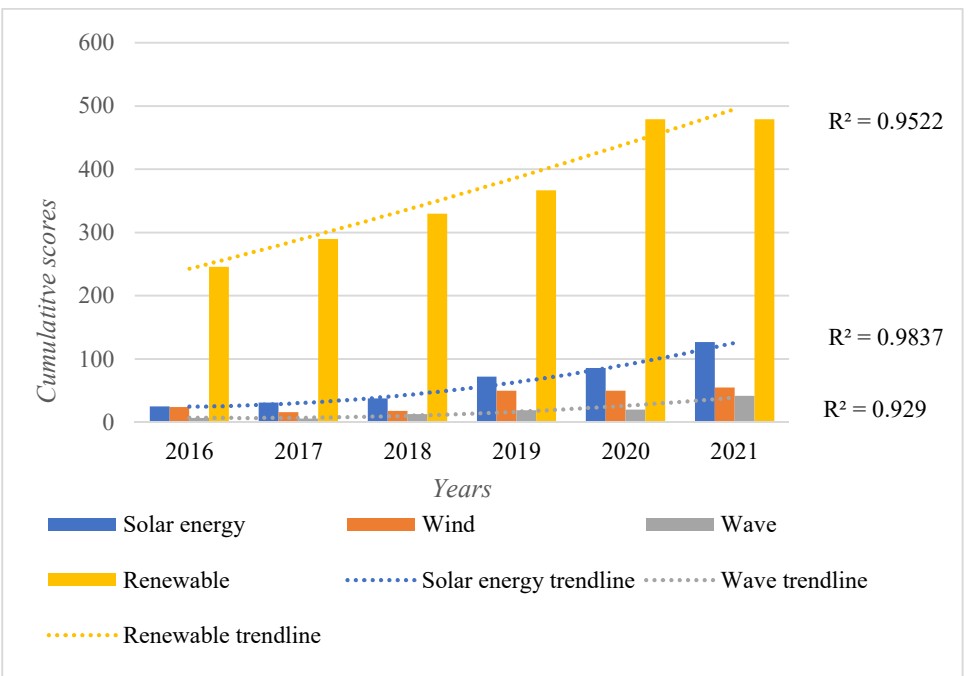

**Figure 7.** Citations on renewable energy sources in the context of energy security education.

Regarding the number of publications, it can be noticed that there is a significant decrease for the terms "renewable", "wind", and "wave", while "solar" energy seems to be gaining interest but does not exceed its previous years' records. Due to the fact that the data did not fit the model well for renewable ($R^2 = 0.3926$), solar energy ($R^2 = 0.5608$), wind ($R^2 = 0.6633$), wave ($R^2 = 0.1846$), trendlines are not presented for these terms.

In the case of citations, there is an increase in all categories. The most-cited topic in energy security education concerns "renewable" sources of energy, and the least-cited issue applies to "wave". The data fit the model for the concepts: "solar energy" ($R^2 = 0.9837$), "renewable" ($R^2 = 0.9522$), and wave ($R^2 = 0.929$), while the fit is poorer for "wind" ($R^2 = 0.7606$).

## 6. Discussion

Increased interest in energy security education, particularly in renewable energy sources, will be triggered by two factors: energy transition [47], along with the technological advancements in the energy sector, and innovations in didactics [48].

Teaching about energy transition is one of the key aspects of energy security education. In many regions, the use of energy sources differs, from nonrenewable to renewable. The use of nonrenewable energy, specifically fossil fuels, which leads to extensive degradation of the environment and accelerates climate change, is associated with negative effects on poor and marginalised communities [49]. These "left behind" societies have a low capacity to deal with environmental hazards and insufficient access to infrastructure to protect themselves from environmental hazards, and prevention services to environmental hazards rarely exist [50]. In these circumstances, countries "should take actions to reduce energy inequalities within and between the regions" [51]. One of the actions focusses on education on energy transition. Such educational efforts should involve a global perspective and be based on reliable sources of data, such as the World Energy Trilemma Index, the report that ranks countries with regard to their capacity to provide affordable, safe, and environmentally sustainable energy infrastructure [52]. It should be noted that education on energy transition also requires knowledge on technological advancements, which apply to novel materials development, new engineering processes, automatisation, and grids for the energy sector. Education for energy security should explore and explain state-of-the-art solutions that can be used for clean energy production.

Within this scope, one of the critical issues is carbon capture, utilisation, and storage (CCUS). According to International Energy Agency's studies, it is one of the four crucial pillars of global energy transformation, together with electrification based on renewable energy sources, bioenergy, and hydrogen [53]. CCUS can reduce emissions, mainly from power plants and large industrial plants, in various ways and reduce harmful emissions by combining them with bioenergy (BECCS) or through direct air capture (DAC) [54]. Carbon dioxide removal technologies can be helpful in specific sectors, including certain industries (especially steel, chemical, and cement), aviation, road transport, and shipping. In addition, CCUS can also provide the possibility of producing low-emission hydrogen based on natural gas or coal ($CO^2$ capture can also be used, for example, to produce clean aviation fuels) [55]. Education on such advancements and their impact on the environment requires collaboration with industry, analysis of case studies concerning state-of-the-art technical solutions, and the generation of novel ideas using, for instance, design thinking methods. It seems that the progress in science and technology is significant, but changes in the technologies used in industry are marginal and mainly result from economic fluctuations. Therefore, the awareness of society and education on energy security is essential to accelerate pro-ecological changes.

Didactic innovations refer to a variety of disciplines, where learning about advancements requires the appropriate choice of didactic tools, form, and methods [56]. In energy security education, the use of online materials on different devices [57], virtual reality, augmented reality [58], advanced systems based on artificial intelligence [59], modelling tools [60], dedicated didactic methods such as simulation [61], challenge-based learning [62], or research-based teaching [63] can be conducive to achieving didactic goals.

The two abovementioned factors seem to be crucial, especially in the process of integrating energy security issues into the formal education system [64].

As the use of energy sources is changing over time and energy security, energy equity, and environmental sustainability are becoming prevalent topics [65], further research may explore the combination of topics: energy security education and sustainable development. Other research topics pivotal for energy security education can relate to interests in the stability and endurance of energy infrastructure [66]. Transformation towards a carbon-neutral energy system employs the use of the Internet of Things and networked devices. This digitalisation poses the main challenge: it makes the critical infrastructure vulnerable to a cyberattack. Such an attack may result in inadvertent disruption [67]. The risk of destruction or disruption of the critical infrastructure by cyberattacks becomes a real threat, especially in the face of the digitalisation and automatisation of energy production, distribution, and use. This topic may be significant for cross-sectional studies on energy security education and for further didactic practice.

From a practical point of view, recognition of the research trends allows for more careful implementation of energy security issues into the curricula. The exemplary topics may concern the sustainable management of natural sources and green energy [68]. Beyond the curricula, it is possible to design dedicated programmes, courses, or events, such as the world climate conferences organised by United Nations agencies [69]. To build awareness in society, it is necessary to strengthen the efforts of actors to provide up-to-date, well-tailored didactic paths to learn about energy security in relation to energy sources.

## 7. Conclusions

This research aimed to identify research trends (reflected by publication activity and citation count of 660 publications from 2016 to 2021) in energy security education. Regarding the concept of "energy security education", the number of publications seemed to decrease, but citations increased significantly. Then, the analysis focused on renewable and nonrenewable energy sources as one of the main challenges for energy security. During the given timeframe, the number of publications on renewable and nonrenewable sources in energy security education tended to decrease (apart from "nuclear power" and "solar energy"; they both seem to have increased since 2020). The number of citations of publications on nonrenewable energy sources has increased for "oil, petroleum", "nonrenewable", and "coal", but for "fossil fuels" and "nuclear power", it dropped. For renewable energy sources, citations increased for all categories. Trendlines were also applied to investigate whether data fit the applied model. A polynomial estimation was used.

For the general term "energy security education", the data fit the model well both for the number of publications and citations. For the analysis of publications on nonrenewable sources of energy in the educational context, trendlines are not presented, as the data did not fit the model well. Regarding citations, a trendline was created only for "coal". In the case of renewable sources of energy, there was also no trendline, but for citations, trendlines were assigned to all categories.

Based on the results, the hypothesis that energy security education trends focus on renewable energy sources was confirmed. Although the tendencies did not differ significantly between the two categories, the overall number of publications on renewable energy sources was higher. There was also an outstanding number of citations for "renewables". Finally, trendlines in the category of "renewable energy sources" were the most accurate. This may imply that nonrenewable energy sources are treated as "contextual" in research activities concerning nonrenewable solutions in energy security education.

**Author Contributions:** Conceptualisation, M.G.-K.; methodology, M.G.-K.; software, M.G.-K.; validation, M.G.-K.; formal analysis, M.G.-K.; investigation, M.G.-K.; resources, M.G.-K., J.S., W.O. and P.M.; data curation, M.G.-K.; writing—original draft preparation, M.G.-K. and W.O.; writing—review and editing, M.G.-K. and P.M.; supervision, P.M. and J.S.; funding acquisition, J.S. All authors have read and agreed to the published version of the manuscript.

**Funding:** This research was funded by the War Studies University Statutory Research Grant, grant number II.1.10.0, 112.

**Institutional Review Board Statement:** Not applicable.

**Informed Consent Statement:** Not applicable.

**Data Availability Statement:** Not applicable.

**Conflicts of Interest:** The authors declare no conflict of interest. The funders had no role in the design of the study; in the collection, analyses, or interpretation of data; in the writing of the manuscript; or in the decision to publish the results.

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
