# Peer review of "Trends in Energy Security Education with a Focus on Renewable and Nonrenewable Sources"

_energies, doi:10.3390/en15041351_

Round 1

Reviewer 1 Report

Paper is well prepared and is suitable for Energies but some minor modification is required. Figure 1 is not clear. A better view is required for this figure. Discussion part should be before Conclusion.

Author Response

Dear Reviewer,

Thank you very much for the prompt and constructive suggestions.

We improved the resolution of the Figure 1 (1780x743). In this way, it looks clearer even when enlarged.

We also placed the Discussion part before Conclusions.

Additionally, we taken into account the suggestions from the second Reviewer. Please find the list of the remaining changes in the attachment.

Kind regards,

Małgorzata Gawlik-Kobylińska

Reviewer 2 Report

The paper brings an original statistical literature survey scoped to energy security education. Critically evaluating the colossal portfolio of papers (title, abstract, keywords), the authors present the mutual proportions among conventional and renewable energy sources for the analysed papers.

I have these professional comments to the manuscript:

1) Avoid writing from the first-person perspective, e.g. do not write “I”, “we”, “our”.

2) Do not attribute single statements to multiple references, i.e. lumped references, instead point to the unique contribution for each cited work and why it is essential.

3) Follow the compulsory structure of the manuscript—no Conclusions before Discussion.

4) Comment regarding Figure 3-7. Each point (number of publications, number of citations, etc.) was reached at a given year (2016, 2017, etc.). As demonstrated by the authors, the points cannot be connected by a full line due to its single score in the year and no continual measurement in time. Just trends can be visualised as correctly presented by trend lines. Thus, delete all the full lines, please.

5) I miss a deeper critical discussion regarding the relationship between academic scientific papers towards industrial situation and development in the energy sector.

  • Does your statistical literature survey follow the development in the industrial energy sector?
  • Can you discuss the literature survey results in the industrial situation?
  • What about CO2 reduction, CCS or CCU technologies and their effect on energy security education?

Based on the information above, I recommend the major revision of the paper.

Author Response

Dear Reviewer,

We are grateful for the constructive remarks; they were very helpful in improving the content of the manuscript. We attached a list of remarks and our replies. 

Due to suggestions of another Reviewer, we enhanced the resolution of Figure 1 and placed the Discussion part before Conclusions.

Kind regards,

Małgorzata Gawlik-Kobylińska

Round 2

Reviewer 2 Report

The paper was improved, I recommend accepting it in the present form.